# Different historical generation intervals in human populations inferred from Neanderthal fragment lengths and mutation signatures

Moisès Coll Macià [1,3 ✉], Laurits Skov[2,3], Benjamin Marco Peter [2] & Mikkel Heide Schierup [1 ✉]

After the main Out-of-Africa event, humans interbred with Neanderthals leaving 1–2% of Neanderthal DNA scattered in small fragments in all non-African genomes today. Here we investigate what can be learned about human demographic processes from the size distribution of these fragments. We observe differences in fragment length across Eurasia with 12% longer fragments in East Asians than West Eurasians. Comparisons between extant populations with ancient samples show that these differences are caused by different rates of decay in length by recombination since the Neanderthal admixture. In concordance, we observe a strong correlation between the average fragment length and the mutation accumulation, similar to what is expected by changing the ages at reproduction as estimated from trio studies. Altogether, our results suggest differences in the generation interval across Eurasia, by up 10–20%, over the past 40,000 years. We use sex-specific mutation signatures to infer whether these changes were driven by shifts in either male or female age at reproduction, or both. We also find that previously reported variation in the mutational spectrum may be largely explained by changes to the generation interval. We conclude that Neanderthal fragment lengths provide unique insight into differences among human populations over recent history.

[1] Bioinformatics Research Centre, Aarhus University, Aarhus C, Denmark. [2] Max Planck Institute for Evolutionary Anthropology, Leipzig, Germany. [3]These authors contributed equally: Moisès Coll Macià, Laurits Skov. ✉email: moicoll@birc.au.dk; mheide@birc.au.dk

If Neanderthal sequences in all non-Africans stem from a single introgression event, then differences in Neandertal fragment length distribution across the world would be indicative of differences in the speed of the recombination clock in among human populations. Assuming a constant number of recombinations per generation, this would then imply differences in the number of generations since the admixture event and consequently differences in generation times among populations. While recent studies point towards a single gene flow event[1], an additional admixture event private to Asians has also been proposed[2–4] because Asian genomes carry larger amounts of Neanderthal sequence compared to European genomes. However, Asian genomes will also have more archaic fragments if a single gene flow common to Eurasians was followed by dilution of Neanderthal content in Europeans, due to subsequent admixture with a population without Neanderthal admixture[1,5].

An independent source of information for estimating differences in generation time is the rate and spectrum of derived alleles accumulating in genomes over a given amount of time[6,7]. Pedigree studies have shown that the yearly mutation rate slightly decreases when the generation time increases because the mutational burst in the germline before puberty represents a high proportion of new mutations in young parents[8]. Moreover, the relative proportion of each mutational type depends on both the paternal and maternal age at reproduction. This has been exploited to estimate differences in generation intervals for males and females between Neanderthals and humans[6].

Here we investigate archaic fragment length distributions among extant non-Africans genomes from the Simons Genome Diversity Project (SGDP)[9] and six high coverage ancient genomes. We report evidence for a single Neanderthal admixture event shared by all Eurasian and American individuals, enabling us to make use of archaic fragment length distributions as a measure of generation intervals since admixture. Differences in estimated generation intervals are mirrored by concordant patterns of mutation accumulation and suggest significant differences in the generation time interval experienced by different Eurasian regions since their splits 40,000 years ago.

## Results

**Neanderthal fragment length distributions differ across Eurasia.** The average archaic fragment lengths in non-African individuals from the SGDP, inferred using the approach of Skov et al.[10], differs across Eurasia and America (Fig. 1a, Methods, S1, Data1_archaicfragments.txt). It presents a clear west-east gradient with the lowest mean fragment length in an individual from the Middle East (S_Jordanian-1, mean = 65.69 kb, SE = 2.49 kb, sd = 72.09 kb, Methods) and the highest in an individual from China (S_Tujia-1, mean = 88.70 kb, SE = 3.29 kb, sd = 110.62 kb, Methods, Supplementary Fig. 5). The pattern is qualitatively very similar when a) median fragment length instead of mean length is used, b) restricting to fragments most closely related to the Vindija Neanderthal genome, the sequenced Neanderthal that is most closely related to the introgressing Neanderthal population[11] or c) only using high-confidence fragments inferred by the model (Supplementary Fig. 2a–c, S2). When individuals are grouped into five main geographical regions, the average archaic fragment length distributions are significantly different ($P$ value < 1e−5, permutation test, Methods) by up to 1.12-fold (Fig. 1b zoom in, Supplementary Table 2). Very similar and significant differences are also found in the independent Human Genome Diversity Project (HGDP)[1] data when comparing more homogeneous populations from each region (Sardinians and Lahu, S5, Data3_HGDParchaicfragments.txt). These five regions also show significant differences in the number of archaic fragments and in the amount of archaic sequence inferred per individual ($P$ value < 1e-5 for both, permutation test, Methods, Fig. 1c, d, Supplementary Table 2), mirroring the mean archaic fragment length distribution patterns. In agreement with previous reports[1,4], we find that East Asians have 1.32-fold more archaic sequence inferred per individual compared to West Eurasians ($P$ value < 1e−5, permutation test, Methods, Fig. 1d, Supplementary Table 2).

We next investigate whether the larger amount of archaic sequence in East Asians is explained by having distinct archaic fragments due to a second Neanderthal admixture. We do this by joining the fragments of the 45 East Asian individuals and comparing them to the joined fragments of a subsample of 45 West Eurasian individuals (Supplementary Fig. 7a, b, Methods, Supplementary Fig. 8). A total of 916,369 kb of the genome is covered by the archaic sequence in East Asia and 866,945 kb in West Eurasia, with 485,255 kb (53 and 56%, respectively) of the archaic sequence overlapping (Fig. 2a, Supplementary Table 6). Thus, as a group, East Asia has only 6% more genomic positions with archaic introgression evidence and does not present the excess of private archaic sequence that would be expected from a private pulse (Methods, S7, see simulation study below). If we further remove fragments with the closest affinity to the sequenced Denisovan (S6), which East Asians are known to possess more of[12], the total sequence covered by archaic fragments is almost identical (East Asia 853,065 kb, West Eurasia 850,028 kb, Supplementary Table 8). When we restrict to fragments with affinity to the Vindija or Altai Neanderthal (S6), East Asia has a 7% higher proportion of the genome covered (East Asia 646,710 kb, West Eurasians 604,518 kb, Supplementary Table 8). We ascribe this latter difference to the fact that shorter fragments in Western Eurasians both make them slightly harder to infer by the Skov et al.[10] approach and less likely to carry single nucleotide polymorphisms (SNPs) that directly classify them as closest to the Vindija Neanderthal.

To compare shared fragments in terms of length, we only consider fragments in East Asian that overlap with regions in the genome of West Eurasians that contain archaic sequence and vice versa (Supplementary Fig. 2d, Supplementary Fig. 7c, Methods). We observe that shared fragments in East Asian individuals are on average 1.13-fold longer than in West Eurasians ($P$ value < 1e−5, permutation test, Methods, Fig. 2b, Supplementary Table 7) as also observed when all fragments were used above.

Based on these observations, we conclude that the vast majority and possibly all of the Neanderthal ancestry in East Asians and West Eurasians stems from the same Neanderthal admixture event, as shown by the group-level analysis. This is supported by simulations that show that an extra Neanderthal admixture into East Asians changes the mean fragment sizes only slightly (2.5%) and causes an excess of private East Asian archaic fragments not observed in our analyses (S7). The 32% greater total amount of archaic sequence in an East Asian compared to a West Eurasian individual on average is primarily due to archaic fragments occurring at higher frequency in East Asians (Fig. 2c, Supplementary Fig. 7d, Supplementary Fig. 8). The shift in frequency is unlikely to have occurred by natural selection acting much more strongly against archaic fragment frequency in West Eurasia, since the purging of Neanderthal introgression is expected to have acted prior to the split of European and Asian populations[13,14]. We consider our observations more compatible with Europeans mixing with a Basal Eurasian population with little or no archaic content diluting the Neanderthal ancestry as has previously suggested from admixture modelling using ancient samples[5]. Such a dilution process would shift their frequency distribution as we observe (Fig. 2c). However, it would have a negligible effect on the average length of Neanderthal fragments, and can therefore not explain the length differences we observe between West Eurasia and East Asia (Fig. 2b).

Differences in the recombination landscape among populations can potentially affect the archaic fragment lengths distribution.

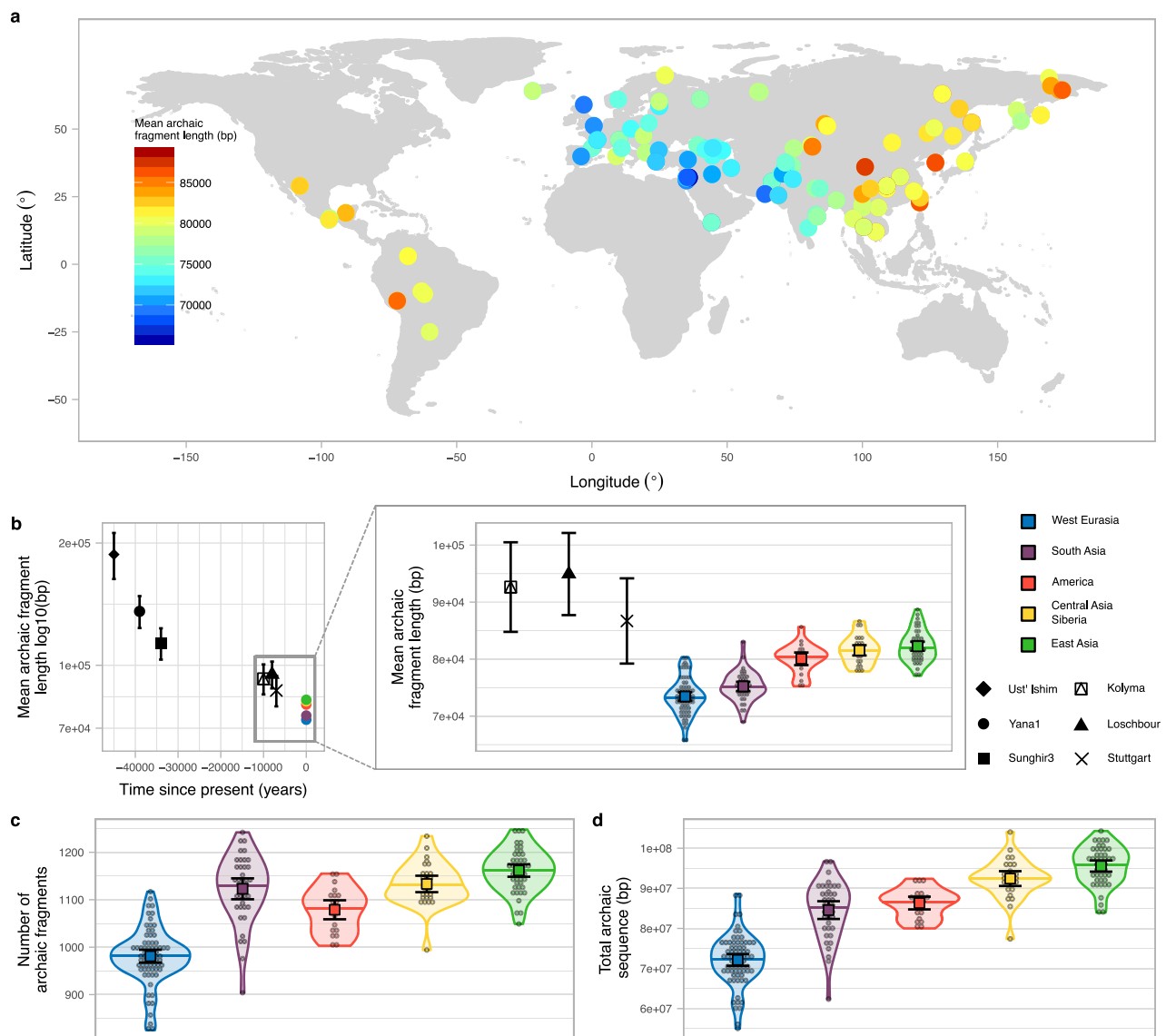

**Fig. 1 Archaic fragment statistics distributions around the world and in ancient samples. a** World map (Methods) showing as circles the samples from SGDP used in this study coloured according to the mean archaic fragment length. **b** Mean archaic fragment length of extant geographical regions and ancient samples. Ust'-Ishim, Yana1, Sunghir3, Kolyma, Loschbour and Stuttgart mean archaic fragment lengths are shown as black points with specific shapes with their corresponding 95%CI as error bars. The sample sizes of fragments for which summary statistics are derived, together with other statistics, are indicated in Supplementary Table 3. The average of the mean archaic fragment length among all individuals in each of the five main regions are shown as points (colour-coded). The zoom-in shows the mean archaic fragment length distribution per region (colour coded) as a violin plot. Individual values are shown as dots. The median is shown as a horizontal line in each violin plot. The mean and its 95%CI of each distribution are shown as a coloured square with their corresponding error bars. The sample sizes of individuals for each region for which summary statistics are derived, together with other statistics, are indicated in Supplementary Table 2. Kolyma, Loschbour and Stuttgart mean fragment lengths are also shown for comparison. **c, d** The number of archaic fragments and the archaic sequence distributions, respectively, per region (colour coded) as violin plot. Individual values are shown as dots. The median is shown as a horizontal line in each violin plot. The mean and its 95%CI of each distribution are shown as a coloured square with their corresponding error bars. The sample sizes of individuals for each region for which summary statistics are derived, together with other statistics, are indicated in Supplementary Table 2.

We use population-specific fine-scale recombination maps[15] to convert the physical length of inferred fragments into recombination lengths (Methods, S4). We find that the differences between Western Eurasia and East Asia are quantitatively very similar to those for the fragments measured in base pairs (Supplementary Figs. 3, S4). This leaves us with a difference in the speed of the recombination clock since the common admixture with Neanderthals as the most likely cause of differences in archaic fragment length distributions. We, therefore, propose that a shift in

generation intervals is the most likely cause of the fragment differences we observe.

Ancient genomes allow us to look at archaic fragment lengths back in time. We called archaic fragments in six high-coverage ancient samples[16–20] dating between 45,000 to 7,000 years ago (Methods, S1, Supplementary Table 1, Supplementary Fig. 1, Supplementary Table 3). As expected, all ancient samples have on average longer archaic fragments than extant populations and samples that date closer to the Neanderthal introgression event

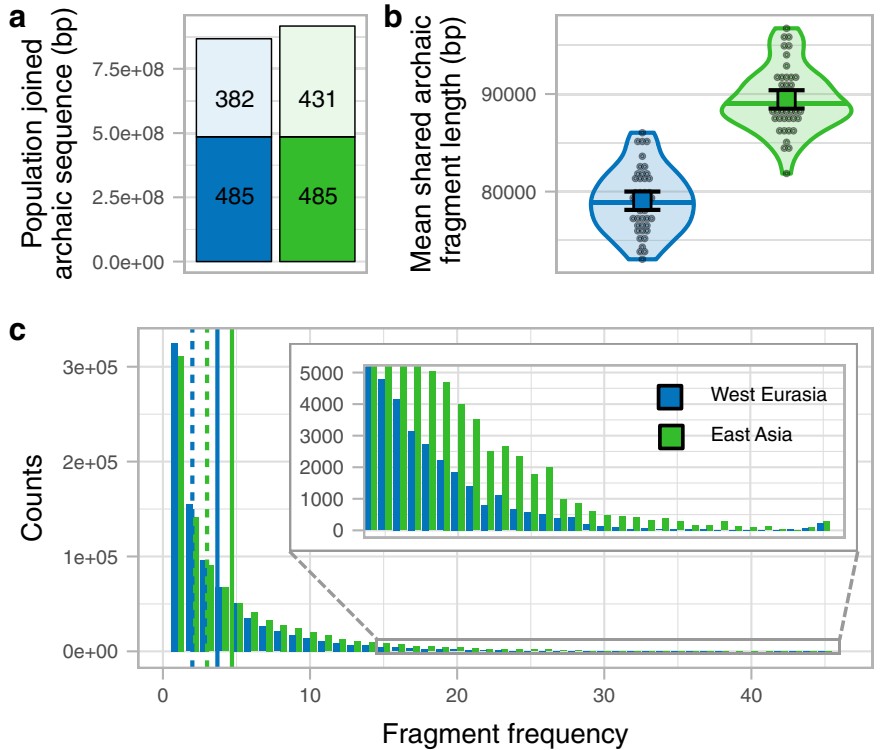

**Fig. 2 West Eurasia and East Asia archaic fragments comparison. a** Joined archaic sequence in both geographic regions (colour coded). The portion of the bar painted in plain colour shows the shared amount between the regions. The rest of the column shows the sequence private of each region. The numbers in each section denote the corresponding archaic sequence in Mb. The sample sizes of individuals for each region for which summary statistics are derived from, together with other statistics, are indicated in Supplementary Table 6. **b** The mean archaic fragment length distributions of individual shared fragments among regions per region (colour coded) as violin plot. Individual values are shown as dots. The median is shown as a horizontal line in each violin plot. The mean and its 95%CI of each distribution are shown as a coloured square with their corresponding error bars. The sample sizes of individuals for each region for which summary statistics are derived from, together with other statistics, are indicated in Supplementary Table 7. **c** The number of 1 kb genomic windows (y-axis) in which an archaic fragment has been found in a certain number of individuals (x-axis) for each region. The insert shows the high-frequency bins. Vertical lines show the mean (plain lines) and median (dashed lines) for each region. The sample sizes of individuals for each region for which summary statistics are derived from, together with other statistics, are indicated in Supplementary Table 7.

have longer mean archaic fragment size inferred (Fig. 1b, Supplementary Fig. 1, Supplementary Table 3, see also Fu et al.[16] and Moorjani et al.[21]). However, Stuttgart—which is a farmer directly related to West Eurasian populations—has an overlapping mean interval with the regions that have the longest fragments (Fig. 1b zoom in). This suggests that East Asian populations, for example, must have experienced similar amounts of recombination than the ancestors of West Eurasians, represented here as the Stuttgart sample. Therefore, the difference between West Eurasians' and East Asians' means correspond to 100–370 generations (assuming an average generation time of 29 years) over the approximately 40,000 years since the split of European and Asian populations[1,22–24].

**Mutations accumulated differently across Eurasia**. The number of de novo mutations (DNM) transmitted to a child depends on the sex and the age of its parents[8]. Thus, a change in generation time during recent human evolutionary history, as suggested above, should leave a detectable pattern in the total number of mutations accumulated. To test this, we estimate the number of derived alleles accumulated in each individual's autosomes since the split of African and non-African populations (Methods, S8, Data2_mutationspectrum.txt). This is done by first removing all derived alleles observed in the sub-Saharan Africa outgroup, excluding those individuals with detectable West Eurasian ancestry[9]. Furthermore, we mask all genomic regions with evidence of archaic introgression in any individual in the study,

since archaic variants would not be found in sub-Saharan genomes and they would affect our results because they accumulated under a different mutational process[6]. Masking those regions also ensures that this analysis is independent of the archaic fragment length analysis above. After these procedures, we are left with ~20% of the callable genome (S8).

Figure 3a shows that the rate of accumulation of derived alleles is significantly different among groups (P value = 3e−4, permutation test, Methods, Supplementary Table 9). West Eurasia has accumulated 1.09% more derived alleles than East Asia (P value = 1.3e−3, permutation test, Methods) since the Out-of-Africa event. However, this difference in the accumulation of derived alleles could only have happened when West Eurasia and East Asia were separated, which is only a part of the time since the Out-of-Africa (S9, Supplementary Fig. 13). If we assume >60,000 years for the Out-of-Africa and a West Eurasia—East Asia split of <40,000 years[1,22–24] (S9), the difference in the rate of derived allele accumulation is at least 60,000/40,000 × 1.09% = 1.64% while West-Eurasia and East Asia were apart (Methods). Using the pedigree-based estimate of the relationships between mean parental age and mutation rate per generation[8] (Methods), we estimate that this difference corresponds to a 2.68 or 3.39 years shorter generation interval in West Eurasia if East Asian mean generation time was 28 or 32 years respectively (Methods, S9). These are lower bounds of the inferred differences in generation intervals since the difference between Out-of-Africa and population split times is minimised.

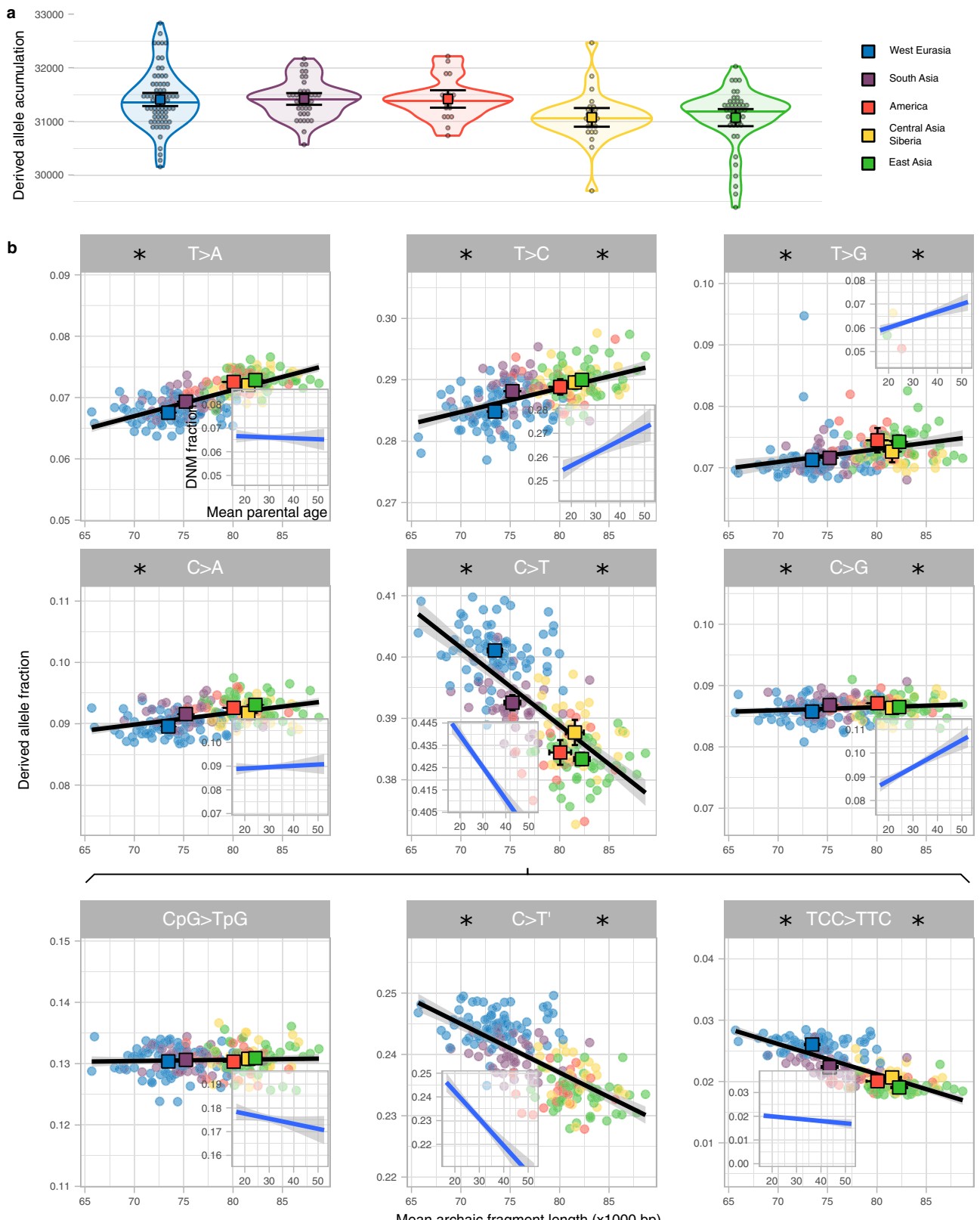

The age of parents at conception, and hence generation time, also impact the frequency of which types of single nucleotide mutations occur[8]. Thus, a shift in generation time is predicted to change the spectrum of new mutations[6,7] and partially explain differences in the mutation spectrum described among human populations[25–27]. We calculate the relative frequencies of the six different types of single nucleotide mutations depending on their ancestral and derived allele (Methods, S8, Supplementary Fig. 12, Supplementary Table 10) and related that to the average Neanderthal fragment length for each individual (Fig. 3b). We observe significant associations with average archaic fragment lengths for all six types (Supplementary Table 11). We further

**Fig. 3 Derived allele accumulation distributions and their mutation spectrum. a** Distribution of the derived allele accumulation (y-axis) per region (colour coded) as violin plot. Individual values are shown as dots. The median is shown as a horizontal line in each violin plot. The mean and its 95%CI of each distribution are shown as a coloured square with their corresponding error bars. The sample sizes of individuals for each region for which summary statistics are derived from, together with other statistics, are indicated in Supplementary Table 9. **b** Correlation between the derived allele proportion (y-axis) with the mean archaic fragment length (x-axis) for nine mutation types. Each dot represents an individual coloured according to the region they belong to. For each region, The mean and its 95%CI of both axes are shown as a coloured square with their corresponding error bars. The sample sizes of individuals for each region for which summary statistics are derived from, together with other statistics, are indicated in Supplementary Table 2 and Supplementary Table 10. Linear regressions (black lines) are shown with their corresponding SE (shaded area). For each mutation, the linear regression and corresponding SE between the fraction of DNM and mean parental age per proband of the deCODE data (Methods, S10) is shown as an insert. Note that the total span of the y-axis is the same for all panels and inserts but centred at the mean value specifically in each panel and insert. Asterisk on the left and right sides of each mutation type indicates that the slope of the linear regression is significantly different from 0 for the SGDP and the deCODE data, respectively (Supplementary Table 11).

subdivide C > T mutations into the following three types: CpG > TpG which present a distinct mutational process[28], TCC > TTC, which is in great excess in European genomes and has been studied as a population-specific mutational signature[25,26] and the rest, denoted as C > T' (Fig. 3b, Supplementary Table 11). We find that the frequency of CpG > TpG transitions depends the least on fragment length.

To investigate whether these correlations could be due to differences in generation time between geographical regions, we reanalyse the proportion of DNM mutation types as a function of mean parental ages in the deCODE trio data set[29] (Fig. 3b inserts, Methods, S10, Supplementary Table 11). Comparing the correlations from the SGDP data with the deCODE data we see a strong correspondence for most mutational types: in all types for which correlations with either data set are significant, the direction of the effects are concordant (Fig. 3b). Moreover, we correlate the slopes estimated in both datasets in order to quantitatively study the similarity of the linear models for each mutation type. We find a relationship close to the 1-to-1 correspondence (slope = 1.058, P value = 1.77e−3, Supplementary Fig. 14) indicative of similar patterns of mutation spectrum variation in the polymorphism data than in the single generation mutation data with estimates of generation time. The deCODE data set has a slight bias towards probands having older fathers than mothers (mean = 2.77 years, sd = 4.25, Supplementary Fig. 15), and this could affect the response of mutation type fraction depending on mean parental age. However, no major change in the correlation coefficients is observed when only probands with similar parental ages are analysed (S10, Supplementary Fig. 16).

Since there is no a priori reason to expect a relationship between archaic fragment lengths and derived allele accumulation, we consider it likely that the same underlying factor has affected both. The general correspondence of these correlations with those expected from DNM studies supports our hypothesis that this causal element is a change in generation interval. More specifically, the matching decreasing correlation with parental age of TCC > TTC mutation indicates that this mutation signature will increase when the mean parental age decreases. Thus a considerable reduction in mean generation time in West Eurasians, as suggested in this study, offers an alternative explanation to the excess of TCC > TTC mutations in that region compared to the rest of the world[26,30].

An increase in the mean generation interval can be due to an increase in paternal or maternal age, or both. Anthropological studies suggest that males have generally been older than females at reproduction, but that the age gap is twice as large in hunter-gatherers compared with sedentary populations[31]. To gain insight into sex-specific changes to generation time intervals we first compare the accumulation of derived mutations between autosomes, which spend the same amount of evolutionary time in both sexes, and X chromosomes, which spend 2/3 of the time in females while 1/3 in males (S11). Thus, an increase of the relative male-to-female generation interval is expected to increase the X chromosome to autosomes (X-to-A) mutation accumulation ratio[32], although other factors such as reproductive variance and changes in population size can also influence the ratio. Figure 4a shows the X-to-A ratio of derived alleles accumulated per base pair (Methods) as a function of the mean archaic fragment length, as mean generation time proxy, for the females in the SGDP data. We observe that the X-to-A ratio is significantly different among regions (P value = 4.4e−4, permutation test, Methods). East Asians have a higher X-to-A ratio compared to American and Central Asia and Siberia, with similar Neanderthal fragment sizes, and higher than West Eurasians, with smaller Neanderthal fragment sizes. This result is compatible with East Asians having a higher mean generation time than West Eurasians primarily due to an increased paternal age over maternal age at reproduction as compared to Americans and Central Asia and Siberia where the age at reproduction of both sexes are inferred to have increased more similarly. However, we acknowledge that the amount of data is still limited for this test and the conclusion is thus preliminary.

Another sex-specific mutation signature are C > G mutations in genomic regions with clustered DNM in old mothers[8,33]. This signature can be explored to compare maternal ages among groups[6]. We estimate the proportion of derived C > G alleles to other derived allele types in these genomic regions and contrasted it to the same ratio for the rest of the genome, for each individual (Methods, S11). When samples are grouped in the 5 main regions, the C > G ratio in DNM clusters differs significantly (P value = 3.1e−3, permutation test, Methods), and increases with increasing Neanderthal fragment length (Fig. 4b). Notably, America has a higher ratio than Central Asia and Siberians for similar Neanderthal fragment lengths, suggesting a relatively larger impact of old mothers on the overall mean generation time throughout their history. This is in line with the X chromosome analysis in that longer generation times in America were more driven by older mothers as compared to older fathers in East Asia with an intermediate increase of both parental ages in Central Asia and Siberia.

Finally, the Y chromosome is also expected to accumulate more derived alleles in populations with younger fathers, similarly to the autosomes, about 0.4–0.5% per year difference in generation time between two populations. We observe a point estimate of 1.19% larger accumulation between West Eurasia and East Asia (S11, Supplementary Fig. 17, Supplementary Table 13) but this is not significant with the limited data available for the Y chromosome (P value = 0.66, permutation test, Methods).

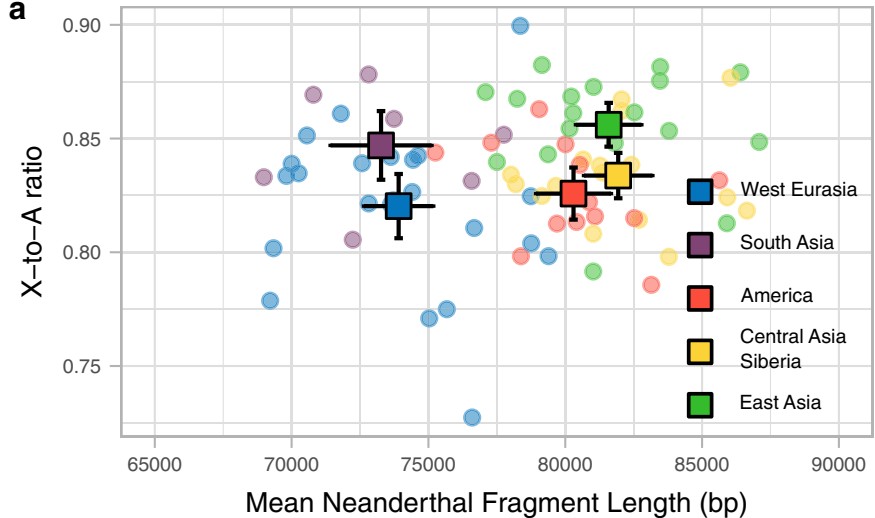

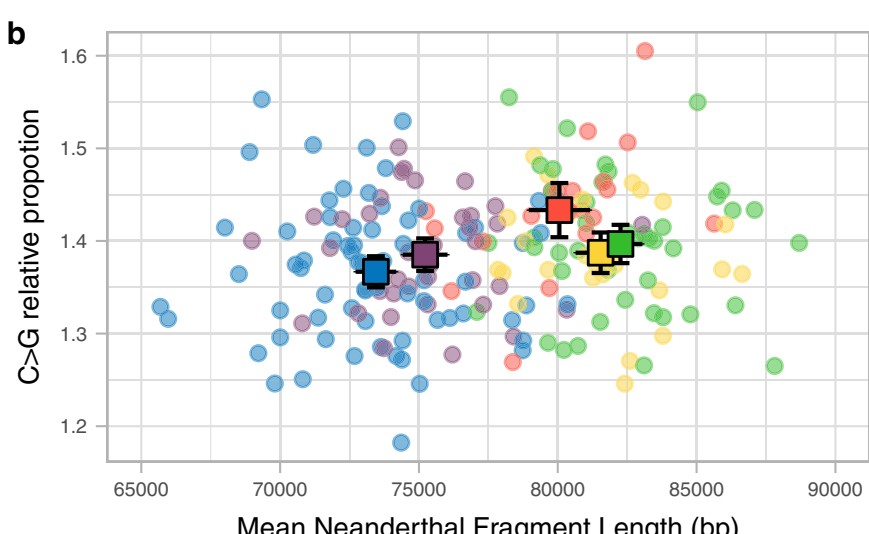

**Fig. 4 Sex-specific mutation patterns. a** Scatterplot of the X-to-A ratio in derived allele accumulation (y-axis, Methods) and the mean archaic fragment length (x-axis) for each region (colour coded). Each dot represents an individual in the corresponding population. The mean for each population for each axis is shown in squares and their 95%CI are denoted by the error bars. Only females were used to produce this plot. The sample sizes of individuals for each region for which summary statistics are derived, together with other statistics, are indicated in Supplementary Table 12. **b** The same as in (**a**), but the ratio between the proportions of C > G derived alleles in cDNM and the rest of the genome (Methods). All samples were used to produce this plot. The sample sizes of individuals for each region for which summary statistics are derived from, together with other statistics, are indicated in Supplementary Table 2 and Supplementary Table 10.

## Discussion

We have shown that the length of Neanderthal fragments in modern human genomes can be used to obtain information about a fundamental demographic parameter, the mean generation interval. We estimate large differences across Eurasian and American groups suggesting stable differences over tens of thousands of years. Our approach depends on the assumption that archaic fragments trace back to a single Neanderthal admixture event shared by all non-African populations, for which we provide further evidence. Consistent with these results, the number of derived mutations accumulated in the geographic regions studied here follow the expectations of the difference in generation time estimated from the fragment lengths. The agreement between the recombination and the mutation clock signatures argues against confounding factors. For example, a potential bias would be expected if the African outgroup, here used to find archaic fragments in the other individuals, had experienced some ancient gene flow from West Eurasia that we

have not been able to detect. Such a scenario would shorten and remove archaic fragments in West Eurasians, explaining the observed gradient. However, it would also decrease the number of derived alleles in West Eurasia compared to East Asia, which is the opposite of what we report.

Differences in generation intervals of the magnitude and duration that we estimate can account for the observed variation in the mutation spectrum of human populations without an underlying change to molecular factors that condition the mutational outcome in each generation, namely mutators and antimutators. An example of this is the increased frequency of the TCC > TTC mutation in West Eurasians. The differences in generation time, inferred here from archaic fragment lengths, explain more than half of the total variation among individuals (adjusted $R^2 = 55.42\%$).

Our results have direct implications for previous investigations of demographic human parameters, which have typically assumed that the generation interval was shared and constant for distinct

human populations. Thus, future investigations should take variation in the generation time under consideration. We do not have an explanation for the underlying causes of large generation interval differences, but it is plausible that a combination of environmental, technological and cultural contexts of those populations affected the age at which parents had their descendants. With an increasing number of sequenced ancient and modern genomes, we anticipate that the approach we present here can be used to obtain a fine-grained picture of shifts in generation interval during the last 40,000 years that can be directly related to changes in population densities, climate, and culture.

## Methods

**Mean and confidence interval calculation**. The mean and its confidence interval (CI) for any statistic are calculated using the mean and standard deviations of the 100,000 bootstrap sampling distribution of the observed statistic. The code to compute them is provided on the GitHub page[34].

**Statistical significance assessment by permutation test**. The statistical significance of a statistic to compare different groups is assessed by contrasting the observed statistic with a non-parametric null distribution. The null distribution is generated by permuting 100,000 times the original data and calculating the statistic in each permutation.

P values are then calculated as the fraction of permutations that yield a value as extreme or more extreme than what is observed in the data. If no such event is observed in all permutations, we considered the fraction to be $<1/100,000 = 1e-5$. The significance level ($\alpha$) in all tests is considered to be 0.05.

To test if there are differences between two groups for a statistic (for example, average archaic fragment length), we subtract the means of each group. In this case, since this test is a two-tailed hypothesis test, we multiply the obtained P value by two. When we test differences for multiple populations, we compute the F statistic.

The code to compute the statistical significance is provided on GitHub page[34].

**Identification of archaic fragments in non-African individuals and ancient samples**. We called archaic fragments in individuals of the SGDP[9] from Eurasian and American regions, 6 ancient modern humans and 4 populations from the HGDP[1] as described in Skov et al.[6,10]—a step by step tutorial is also available at https://github.com/LauritsSkov/Introgression-detection.

In short, the method first removes a set of variants (SNPs) that are present in an outgroup with no presumed archaic admixture (Sub-Saharan African populations) from the samples in which we want to detect archaic fragments (non-Africans). Then, taking into account window-specific mutation rate and callability, the method classifies non-overlapping windows into archaic ancestry and non-archaic ancestry depending on the derived allele density.

More detailed information about the specifics on calling archaic fragments in each of the datasets can be found in S1 and S5.

**World map plots**. The background map was obtained using the R function borders, part of the ggplot2 library, which retrieves maps with the map function from the maps library. The latter imports a 1:50 m scale world map raster that is publicly available from the Natural Earth project.

**Archaic fragment physical length conversion to genetic distance using population-specific recombination maps**. The mean archaic fragment length differs among the 5 geographical regions of the SGDP data (main text). To test if population-specific recombination maps can account for the differences in archaic fragment sizes that we observe, we transform the fragment lengths from physical units (base pairs, bp) to genetic units (centiMorgans, cM) using the population-specific recombination maps inferred by pyrho[15] on the 1000 Genomes Project (1KGP) and compare the resulting distributions among regions.

We download the recombination maps for populations of the 1KGP[35] from: https://github.com/popgenmethods/pyrho#human-recombination-maps. Supplementary Table 4 shows the subset of populations analysed in this study that have representation in 1KGP, and thus, recombination maps that we can use.

We note that, for different populations, chromosomes have different genetic lengths in the downloaded maps; in general, East Asian populations tend to have inferred longer chromosomes. If we applied these maps directly, it would result in East Asian archaic fragments being even longer compared to the other populations in genetic lengths than in physical lengths. Since there is no reason to believe that chromosome genetic map lengths are different among populations, we scaled recombination rates such that all populations have the same total length for every chromosome to compute the fragment sizes in genetic distances.

For each chromosome ($c$), we first scale the recombination rates ($r$) of every population ($p$) by the length ($l$) of the chromosome of the population with the

shortest size ($q$)

$$r'_{pc} = r_{pc} * \frac{l_{pc}}{l_{qc}}$$

Then, with the updated rates ($r'$), we compute the mean recombination rate ($\bar{r}$) of an archaic fragment ($f$) by intersecting it with the recombination blocks ($k$) and computing the mean, weighting each block's recombination rate by the ratio between the recombination block overlap with the archaic fragment and the total archaic fragment length ($w$)

$$\bar{r}_f = \sum_{i}^{k_f} r'_{fi} * w_{fi}$$

Finally, we converted the length of every fragment ($s$) in physical units (bp) to genetic units (cM) using the mean recombination rate ($\bar{r}$) for each fragment

$$s(cM) = s(bp) * 10^{-6}(Mb/bp) * \bar{r}(cM/Mb)$$

Supplementary Fig. 3 shows the mean archaic fragment length distribution in physical and scaled genetic distances. Compared to the physical length distributions, genetic length distributions have overall less dispersion within regions and greater difference between regions. On average, East Asians have 1.16 times longer fragments than West Eurasia in genetic length units, which is a greater difference than the 1.10 times if the physical length is instead compared. Nonetheless, in both cases, the differences among regions are statistically significant (genetic length P value $< 1e-5$, physical length P value $= 4e-5$, permutation test, Methods). This is due to the good correlation between the two measures of the fragment length for all individuals assessed (Supplementary Fig. 4).

We conclude that differences in genetic maps among populations cannot account for the differences in fragment length among regions described in this study.

**West Eurasia and East Asia fragment comparisons of archaic fragment genomic coverage**. In the main text, we compare fragments in West Eurasians and East Asians. The more individuals used to recover archaic fragments, the more undiscovered fragments can be found[6]. Thus, the imbalance in the number of individuals in each region in the SGDP data (71 West Eurasians and 45 East Asians) can potentially affect any comparison between the two regions. Therefore, we downsample the number of individuals used in West Eurasians to 45 randomly chosen individuals to make comparisons fair.

First, we join all overlapping fragments for each region, hereby "joined region fragments" (Supplementary Fig. 7). To do that, we used bedtools software[36] (version 2.30.0) with the following command:
bedtools merge -i ind1_regx.bed ind2_regx.bed … indN_regx.bed > joined_regx.bed where x denotes either West Eurasia or East Asia regions and N denotes the number of individuals in the corresponding region.

Then, we compared how much archaic sequence the two regions share (Supplementary Fig. 7). For that, we call the intersect between the two joined sets of fragments. We refer to it as the "shared joined region sequence". We use the following command:
bedtools intersect -a joined_regx.bed -b joined_regy.bed > shared_joined.bed where x denotes either West Eurasia or East Asia and y denotes the other region different than x.

It follows that the rest of the fragments not included in this set are the "private joined region sequence".

The amount of sequence for shared, private and total joined region fragments are provided in Supplementary Table 6.

For each individual, we classified the fragments as shared depending upon if there was an overlapping fragment in the other joined region fragments (Supplementary Fig. 7). We name these fragments as "shared individual fragments". To get them, we ran the following command:
bedtools intersect -u -a indn_regx.bed -b joined_regy.bed > shared_indn_regx.bed

It follows that the rest of the fragments not included in this set are the "private individual fragments".

Summary statistics for shared and private individual fragments are provided in Supplementary Table 7.

Finally, we calculated the number of individuals that have an overlapping archaic fragment in a certain 1 kb window in the genome. This way, we calculate the "archaic frequency". For that, we first divided each fragment in the joined region fragments into 1 kb segments (joined_regx_1kb.bed). Then, we counted the number of individuals with an overlapping archaic fragment for each 1 kb segment with the following command:
bedtools intersect -c -a joined_regx_1kb.bed -b ind1_regx.bed ind2_regx.bed … indN_regx.bed > freq_regx.bed

Supplementary Fig. 8 shows a summary of the joined region fragments, shared joined region sequence and the archaic frequency for each region.

**Simulations**. We simulate using msprime[37] whole genomes for two demographic scenarios: a scenario with a single Neanderthal pulse to the common ancestors of East Asians and West Eurasians (One Pulse) and another one with an additional and private pulse to East Asians (Two Pulses, S7). The parameters for both

scenarios are shown in Supplementary Fig. 9 and some are indicated in the following list:

—Mutation rate = 1.25e−8
—Recombination map = HapMap recombination maps[38] downloaded from http://bochet.gcc.biostat.washington.edu/beagle/genetic_maps/
—Generation time (years) = 29

The reasons why we chose such parameters are explained in S7.

For each scenario, we performed ten replicates of whole genomes in order to obtain an estimate of the variance in the statistics analysed.

In each simulation, we sample 500 individuals of the Africa group as an outgroup and 50 of each West Eurasia and East Asia. Variant genomic positions are then transformed to a discrete space by truncating the given float value into an integer value and positions with multiple variants are removed. To simulate a similar genomic callability profile in the simulated data, we disregarded variants that fall into non-callable positions of the files used in S1. We then called archaic fragments following a similar methodology to the one described in the Methods section and in S1.

The analyses of these simulations are detailed in S7.

**Derived alleles call outside regions with evidence of archaic introgression and acquired after the Out-of-Africa in SGDP samples**. We retrieved the genotypes of all polymorphic loci for each individual in the 5 main regions and African samples using the `cpoly` script from the `Ctools` software (version 1500)[9] for chromosomes 1–22. In the parameter file, we specified the minimum quality to be 1 (as recommended by Mallick et al.[9]) and alleles to be polarised with the chimpanzee reference genome (PanTro2) provided with the SGDP data.

Next, we masked repetitive regions and regions of the genome in which there is some evidence of archaic introgression. This is because repetitive regions might be enriched with sequencing errors and also because Neanderthals had a different mutation profile than modern humans[6]. Moreover, by removing these regions, we will base the mutation analysis on regions of the genome that we haven't explored in the archaic fragment length part of the study. Thus, the tests are going to be independent of each other. Further details on how this was performed for this study are explained in S8.

Other filters on the SNP level were imposed for each polymorphism such as retaining only biallelic SNPs or the derived allele for that locus to not be segregating in the African outgroup. Further details on the filters applied are outlined in S8.

Homozygous locus for the derived allele count as 2 mutations and heterozygous sites count as 1 for a given individual. The distribution of derived allele accumulation per region is shown in Fig. 3 and the mean derived allele accumulation counts per region are provided in Supplementary Table 9.

We classified loci in different mutation types depending on the derived allele nucleotide, the ancestral allele nucleotide and their 5′ and 3′ nucleotide context, further explained in S8. Data2_mutationspectrum.txt provides the resulting counts of each individual for each mutation type in each chromosome.

**Estimation of the different parental generation time in West Eurasia and East Asia**. To transform the excess of derived allele accumulation in West Eurasia compared to East Asian regions after the Out of Africa we assume that the African exodus happened 60,000 years ago and that the split between West Eurasia and East Asia populations happened 40,000 years ago. The reasons why those conservative dates for this analysis are assumed are detailed in S9. Because the amount of derived alleles is proportional to the Out of Africa event, but the excess could only be accumulated after the split of both Eurasian populations, the per cent excess between the derived allele accumulation (1.09%) must be scaled up according to these times

$$1.09\% \cdot \frac{60,000}{40,000} = 1.64\%$$

In Jónsson et al.[8], a Poisson regression is derived for the number of mutations transmitted in each generation from trio data for each parental lineage depending on their age at reproduction

$$\hat{\mu}_{f.g} = 6.05 + 1.51 a_f \tag{1}$$

$$\hat{\mu}_{m.g} = 3.61 + 0.37 a_m \tag{2}$$

Where subscripts $f$ and $m$ denote paternal and maternal respectively, $\hat{\mu}$ is the estimation of the mean mutation rate per generation $(g)$ and $a$ is the mean parental age. Thus, assuming the same mean parental age for both progenitors $(a_f = a_m = a)$ we get that the total mutation rate per generation is calculated by Eq. (3) and the yearly $(y)$ rate by Eq. (4).

$$\hat{\mu}_g = \hat{\mu}_{f.g} + \hat{\mu}_{m.g} = 9.66 + 1.88a \tag{3}$$

$$\hat{\mu}_y = \hat{\mu}_g / a \tag{4}$$

Then, to compare the mutation rate per year in two different populations ($x$ and $z$) with different mean parental ages, we get that

$$\frac{\hat{\mu}_{yx}}{\hat{\mu}_{yz}} = \frac{\frac{9.66}{a_x} + 1.88}{\frac{9.66}{a_z} + 1.88} \tag{5}$$

The number of derived alleles accumulated in a genome during a period of time $(d)$ depends on the mutation rate per year and the time span $(T)$

$$d = \mu_y T \tag{6}$$

However, the ratio of $d$ between the two populations will only depend on their mutation rate because $T$ has been the same for both.

$$\frac{d_x}{d_y} = \frac{\hat{\mu}_{gx}}{\hat{\mu}_{gy}} \tag{7}$$

Thus, we can estimate the $a_x$ if $a_z$ and the $d_x/d_z$ are known

$$\frac{\hat{\mu}_{gx}}{\hat{\mu}_{gz}} = \frac{d_x}{d_z} = \frac{\frac{9.66}{a_x} + 1.88}{\frac{9.66}{a_z} + 1.88}$$

$$a_x = \frac{9.66}{\frac{d_x}{d_z}\left(\frac{9.66}{a_z} + 1.88\right) - 1.88} \tag{8}$$

In this study, we find that the ratio of the mean derived allele accumulation in West Eurasia (WE) vs. East Asia (EA, $d_{WE}/d_{EA}$) is 1.0164 (1.64%). With the formula (8), we check for reasonable $a_{EA}$ values between 28 and 32 years and found that the values of $a_{WE}$ ranged between 25.32 and 28.59 respectively. Thus, we estimate that generation times in East Asians have been 2.68–3.39 years longer than in West Eurasians since the split of the two populations.

**Mutation spectrum correlation with mean parental age**. We compare the mutational patterns of DNM depending on parental age in trio studies[8,29] (deCODE data set) with the differences in mutation spectrum of extant populations with the mean archaic fragment length as a proxy of mean generation time (SGDP data set).

Similarly to Jónsson et al.[8], we study these correlations as linear models following the general formula

$$\text{Mutation fraction} = \alpha + \beta \,(\text{Parental } age \; estimate)$$

Where $\alpha$ is the intercept and $\beta$ is the slope of the linear regression estimated.

For the SGDP data set, we classified the derived alleles found in the autosomes of each individual into six mutation types depending on the ancestral and derived allele as explained in the Methods section and in S8. C > T mutations were also classified into three subtypes: TCC > TTC, CpG > TpG and the rest (C > T'). In total, we divide all mutations into nine types. In order to obtain the fraction of each mutation type per individual, we divided the number of each mutation type by the total amount of derived alleles. C > T mutations are duplicated since we subdivide them into three extra categories (TCC > TTC, CpG > TpG and C > T'). Thus, the total number of derived alleles do not consider these three types. We correlated the fraction of derived alleles of each type with the mean archaic fragment length as a proxy of mean generation time (Fig. 3b). We obtained the linear model of such correlation for each mutation type using the following R function (Supplementary Table 11).

`lm(mutation_fraction~mean_fragment_length)`

For the deCODE data set, we downloaded the set of DNM called in Halldorsson et al.[29] and the additional proband information from the supplementary data provided in the publication ($n = 2976$ trios for which 200,435 DNM). We join both in order to compute the mean parental age for each DNM for each proband. Indels are filtered out. Following the methodology in a similar test in Jónsson et al.[8], we aggregate all mutation counts for each of the nine types of all probands with the same mean parental age. We then compute the fraction of each mutation type. In other words, for each mutation type and mean parental age we have a single mutation fraction value. Those data points that were obtained aggregating information from less than 2 probands were discarded. We obtained linear models for each mutation type using the following R function (Supplementary Table 11).

`lm(mutation_fraction~mean_parental_age, weights = n_probands)`

The correlations between the slopes of both data sets are shown in Supplementary Fig. 14.

**Sex-specific mutational patterns**. The X-to-A ratio is obtained by first calculating the number of derived alleles in the autosomes and X chromosomes of the females of the SGDP data (Supplementary Table 12), as described in the Methods section and in S8 (included in Data2_mutationspectrum.txt). Then, the ratio is computed as

$$\frac{d_X}{L_X} \Big/ \frac{d_A}{L_A}$$

where $d$ denotes the number of derived alleles, $L$ the number of callable base pairs in either $X$ (X chromosome) or $A$ (autosomes).

To compute the C > G ratio between DNM-clustered regions (cDNM) and the rest of the genome (non-cDNM), we compute the number of derived alleles that are C > G and non-C > G along the genome in windows of 1 Mb. We join this information with the annotation of 1Mb-window of the genome as cDNM or non-cDNM provided in Jónsson et al.[8]. Then, for each individual we compute the following

$$p = \frac{d_{C>G}}{d_{non-C>G}}$$

where $d$ denotes the number of derived alleles of C > G or non-C > G. Thus, $p$ is the ratio between the two quantities. Then, to compare this ratio between cDNM and non-cDNM regions we compute the mean $p$ ($\bar{p}$) over all regions and compute the following ratio:

$$r = \frac{\bar{p}_{C>G}}{\bar{p}_{non-C>G}}$$

If $r = 1$, it indicates that the C > G enrichment is similar in cDNM regions compared to the rest of the genome. If $r > 1$, then there is an excess and if $r < 1$, then there is a depletion.

## Data availability

The archaic fragments and their basic statistics for the SGDP samples and ancient samples are provided in the Source Data file Data1_archaicfragments.txt. The counts of the 96 mutation types per individual per chromosome are provided in the Source Data file Data2_mutationspectrum.txt. The archaic fragments and their basic statistics for the HGDP samples are provided in the Source Data file Data3_HGDParchaichfragments.txt (S12). All three Source Data files are provided with this paper. All data used in this study is publicly available. We used variant calls from individuals whole genome sequenced provided by the Simons Genome Diversity Project[9] (https://reichdata.hms.harvard.edu/pub/datasets/sgdp/), the 1000 Genomes Project[35] (https://www.internationalgenome.org/) and the Human Genome Diversity Project[1] (ftp://ngs.sanger.ac.uk/production/hgdp/). We used the variant calls of the archaic Vindija Neanderthal[11], Altai Neanderthal[39] and the Denisova[40] (https://www.eva.mpg.de/genetics/genome-projects/). Finally, the sequenced reads from genomes of the following ancient modern humans deposited in ENA or SRA repositories were also analysed: Ust'-Ishim[16] (PRJEB6622), Yana1[17] (PRJEB29700 and PRJEB26336), Sunghir3[18] (PRJEB22592), Anzick1[19] (SRX381032), Kolyma[17] (PRJEB29700 and PRJEB26336), Loschbour[20] (PRJEB6272) and Stuttgart[20] (PRJEB6272).

## Code availability

The scripts coded to produce data and tables, perform statistical analysis and plot figures for this manuscript are accessible on Github (https://github.com/MoiColl/TheGenerationTime Project)[34]. The scripts provided in this repository are licensed under the MIT License.

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

## Acknowledgements

We thank Felix Riede for advice on anthropological interpretations and Matthew Hurles for suggesting to contrast derived allele accumulation between X and autosomes. We thank Juraj Bergman and Marjolaine Rouselle for all discussions about the consequences of the Neanderthal dilution scenario in West Eurasian populations. We thank Priya Moorjani for commenting on the study and providing feedback and giving insightful suggestions. The study was supported by grants NNF18OC0031004 from the Novo Nordisk Foundation and 6108-00385 from the Research Council of Independent Research to M.H.S.

## Author contributions

M.C.M., L.S. and M.H.S. designed the study. M.C.M. and L.S. created the methods to assess the data and, with M.H.S. analysed the results with input from B.M.P. M.C.M., L.S. and M.H.S. wrote the manuscript with comments from B.M.P.

## Competing interests

The authors declare no competing interests.
