## [Peer Review File · Nature Communications]

Different historical generation intervals in human populations inferred from Neanderthal fragment lengths and mutation signaturesREVIEWER COMMENTS

Reviewer #1 (Remarks to the Author):

Macià et al. find evidence that Neanderthal introgression into non-African genomes arose from a single event and that the length of introgressed sequences is greater in East Asians/Americans than in West Eurasia. This is interpreted as evidence for a longer generation time in East Asia (and hence fewer opportunities for recombination). Supportive evidence for this conclusion is found in the sex-specific distribution of the derived alleles. These are fascinating questions of broad general interest.

I have one major conceptual question. The lower frequency of Neanderthal alleles in West Eurasia is interpreted as evidence for dilution by mixing with a population with no Neanderthal ancestry. At lines 120–121 it is claimed that “such a dilution process should have a negligible effect on the Neanderthal fragment lengths observed today”. I would like to see more justification of this conclusion. The effect of dilution is to decrease the frequency of Neanderthal fragments and therefore increase the frequency with which Neanderthal fragments occur in heterozygotes rather than homozygotes. This would increase the rate at which fragments were broken up by recombination and would therefore contribute to a shorter fragment length in West Eurasia. The effect may be small but the authors should address whether this effect of dilution could account for some variation in fragment lengths.

David Haig

Reviewer #2 (Remarks to the Author):

By leveraging the distribution of Neanderthal ancestry fragments and patterns of mutation accumulation in Europeans and East Asians, the authors show there are significant differences in the human generation interval across non-African groups. This is an innovative idea and a very creative way to learn about historical generation time, a critical parameter in evolutionary studies. The analysis performed is compelling, but I have some concerns about the model and some results that I detail below.

Main comments

1. The key observation the authors focus on is that Europeans and East Asians differ in the % and the length of Neanderthal ancestry tracts. The author suggest the underlying model is that there was a single pulse of gene flow from Neanderthals into the ancestors of modern humans and the observed differences are due to differences in the rate of recombination clock driven by generation time. While feasible, more work needs to be done to justify this model and reject other possible reasons.

a) Single pulse of Neanderthal gene flow: While some studies support this model, there are other studies that support a model of multiple gene flow events (e.g., Vernot et al. 2016). There is also evidence for multiple pulses in ancient DNA samples such as Oase and Ust' Ishim (Moorjani et al. 2016) where the dating is more direct and based on fewer assumptions. Also given that the authors find that only half of the total Neanderthal ancestry segments are shared between East Asians and Europeans, I wonder if a model of multiple gene flow events can be rejected. Worth simulating this to test the predictions and assess whether the model of single or multiple pulses provides a better fit to the data.

b) Differences in recombination rates across Europeans and East Asians: One key reason the authors do not consider is that the differences in recombination rate across Europeans and East Asians could underlie the differences in the Neanderthal ancestry tract lengths across these groups. There is evidence that as recombination evolves quickly due to Prdm9 and the fine scale recombination map differs between Europeans and East Asians. It would be useful if the authors can explore how changes in recombination rate would impact their results, perhaps using new population specific maps for Europeans and East Asians (e.g., Spence and Song, 2019).

c) Selection: The authors reject the possibility that selection could be driving the difference between East Asians and Europeans. However, the observation that East Asians have Neanderthal ancestry segments at higher frequency could be in part due to selection. It would be useful to perform some simulations to test models with selection– both positive and negative selection could contribute to the observed differences across populations.

2. Choice of samples: The authors use SGDP dataset and compare Neanderthal ancestry patterns across West Eurasians and East Asians. However, West Eurasians themselves are a diverse group including European, Central Asian and Middle Eastern groups together – populations that likely have very different generation times and recombination rates. It would be useful to repeat the analysis with more homogenous groups. If larger sample sizes are needed then other datasets like 1000G or high coverage genomes from HGDP (Bergstorm et al. 2020) maybe useful. This will help to interpret the source of the differences more clearly.

3. Inconsistency in signals across populations

The authors show that decrease in Neanderthal ancestry tract length is correlated to increase in derived alleles in Europeans. This makes sense. However, South Asians have similar % of derived allele as West Eurasians but fragment lengths are longer (Table S3 and S6). This is contradictory to their model predictions. Could there any technical issue impacting this result?

4. Figure 3 - For each mutation type, the authors show slopes for two models - correlation in derived allele accumulation and Neanderthal ancestry fragment length and parental age effects. They suggest there is a strong correspondence in the slopes across the two models. However, this analysis is very qualitative and in particular, these slopes look fairly different for many mutation types. It would be useful if the authors can provide quantitative estimates of the similarities and differences for each mutation type.

5. TCC->TCT

It is interesting that the authors find that differences in generation time contribute to the increase in TCC->TCT mutations in Europeans. Could the authors provide some quantitative estimates of what proportion of this difference is explained by generation time vs. other factors?

6. Direct estimate of generation time

The authors provide indirect estimates of generation time differences between Europeans and East Asians. Could they relate the N fragment length distributions and mutation patterns to provide a direct estimate of generation time?

7. Sex-specific changes in generation time

The authors use the enrichment of C>G mutations to compare the maternal generation time across populations. This is an interesting idea. However, the analysis seems very tentative as the ratio seems fairly dispersed within and across populations. Moreover, X/A is impacted by the sex-specific generation time though the Neanderthal ancestry tract lengths on the autosomes depend on the sex-averaged generation time. Further, some of the populations like Americas that show an increased ratio are admixed (with Native American, European or African ancestry) and so it is not clear how to interpret these results -- which ancestry group has a long maternal generation time. A more clear quantitative model is needed here to justify the conclusions.

Minor comments:

1. Specificity and sensitivity of Neanderthal ancestry detection HMM. The authors state that this method has lower sensitivity to detect shorter segments. Could this impact the differences across populations? Would be useful if authors can provide information about the specificity and sensitivity of the method as a function of ancestry tract lengths.

2. On page 2 of results, authors claim that after excluding Denisova fragments the total sequence covered by archaic fragments is almost identical. Does this not suggest that the 7% difference seen earlier is driven by Denisovans in large part?

3. The authors use parental age effects estimated in Europeans and compare them to N fragments across populations. It would be useful if the authors can mention this point in the discussion or results, as these mutation parameters and parental age effects may themselves differ across human populations.

4. Difference in generation time.

The difference in generation time of 5 years between Europeans and East Asians seems tenuous and depends on the timing of the OOA migration. This has been estimated to be between 50,000-100,000 years and is closer to 100,000 based on the pedigree mutation rate as used for other analysis. See Schiffels & Durbin 2014 or Spiedel et al. 2020.

Reviewer #3 (Remarks to the Author):

This article aims at inferring ancient socio-biological patterns (age of mothers and fathers) using genetic data of 2 two types : Neandertal inherited fragment length (that provide insights on the recombination rate) and number of derived alleles (that depends on the number of de novo mutations).

The article is well-written and provides convincing arguments. It also enlightens the process of admixture between Sapiens and Neandertal. However, I have two major concerns.

1) The authors use the mean archaic fragment length of 3 ancient genomes to infer the differences in the number of generations since their split between Asian and European (l. 133-l. 142). However, these genomes are either very ancient (Ust-Ishim, 45 ky cal BP) or quite recent (Stuttgart 7 ky cal BP and Loschbourg 8ky cal BP), thus providing a poor estimation of the decrease of the fragment length through time. I suggest that the authors add a few genomes of west- Eurasian ancestry to their data set, such as Kostenki14 (coverage: 16.1 X, age : 37 ky cal BP) ; Kotias (coverage : 12.2 X, age : 9.5 ky cal BP) or Sunghir (coverage : 10.75 X, age : 34 ky cal BP). It would also be interesting to compare these ancient genomes to Anzick-1, from the Clovis Culture in North America (coverage: 14.4 X, age : 12.5 ky cal BP), or to Yana1

and Kolyma 1 from paleolithic Siberian population (coverage respectively : 25 X and 14.3X, age respectively : 9.7 ky cal BP and 31.6 ky cal BP). The authors should also take into account their lack of precision in the estimation of the “missing recombinations” in Asian and moderate their conclusions (l. 138 – l.142)

2) On the l. 200-l. 201, the authors suggest that archaic fragment length and derived alleles accumulation are not related. On the contrary, mutation rate depends also on the GC-biased conversion during recombination. This phenomenon increases the fixation of C and G alleles compared to T and A. GC-biased gene conversion is linked in human to the activity of PRDM9 protein. Different alleles of PRDM9 exist in the human species and some affect recombination rate (for instance between European and African : <https://advances.sciencemag.org/content/5/10/eaaw9206>). I wonder if this phenomenon could affect the authors results on recombination and mutation rate.

I have a few other observations

3) The authors observe that the Skov et al. method has difficulties to identify very short fragments (l. 100). What is the threshold? could that change the mean/median length of the fragments for both populations (for instance with a higher number of short fragments in Asian genomes)?

4) The authors suggest that only one admixture event occurs in the European/East Asian ancestors (l. 111-112). However, the homogeneity of Neandertal alleles in European and East Asian genomes do not rule out several admixture events with only one homogeneous Neandertal population. Indeed, it seems that Neandertal groups were highly homogeneous on a genetic point of view. If ancestors of East Asian populations have experienced more admixture events/more recent admixture event with this Neandertal population, it could explain the higher percentage of Neandertal alleles and the longer Neandertal fragments in East Asian genomes.

5) In line with others, the authors consider that Neandertal ancestry may have been diluted by gene flow from Basal Eurasian with little/no archaic alleles. In the Skov et al. 2018 paper, it is suggested that “We note that other types of population structure, for example involving continual gene flow, could also create signals under our model.” Could this gene flow impair identification of archaic fragments?

6) In the Fig.4, it seems that East Asian populations have a higher paternal age at conception than other Asian populations (Fig.4.a) AND a higher maternal age than Central Asian/Siberian population. But this difference does not reflect on the mean Neandertal fragment length. Or, to rephrase, Central Asian/Siberian populations do not appear significantly different from West Eurasian populations for paternal and maternal age but have significant Neandertal fragment length. How do you explain this?

7) Last remark: the authors write that this difference in the generation time may be explain by agriculture that decrease generation time (in Western Eurasian). But this difference is already seen in Stuttgart and Loschbour (just before/after Neolithic) ; and the onset of agriculture is more or less the same between East Asian and Western Eurasia. Thus, I do not think this can be the cause of the difference.

8) Overall, the text is carefully written and pleasant to read. Figures are very clear, useful and beautifully done. My main point on the wording is that the aim of the article does not appear clearly at the beginning: both the introduction and the abstract are focused on the admixture between Neandertal and Sapiens, which is in fact only a tool for studying a Sapiens-related phenomenon.

9) In the bibliography, there is a few typos for the following articles: (2) Bergström et al. ; (9) Skov et al. ; (16) Fu et al. ; (27) Fenner et al.

Also a typo l. 150 “non Afrcan populations”

To conclude, I consider this paper as very interesting and worth of Nature Communication, since the comments and concerns I expose are correctly addressed.

Point-by-point response to reviewers' comments

Reviewer #1 (Remarks to the Author):

Macià et al. find evidence that Neanderthal introgression into non-African genomes arose from a single event and that the length of introgressed sequences is greater in East Asians/Americans than in West Eurasia. This is interpreted as evidence for a longer generation time in East Asia (and hence fewer opportunities for recombination). Supportive evidence for this conclusion is found in the sex-specific distribution of the derived alleles. These are fascinating questions of broad general interest.

I have one major conceptual question. The lower frequency of Neanderthal alleles in West Eurasia is interpreted as evidence for dilution by mixing with a population with no Neanderthal ancestry. At lines 120–121 it is claimed that “such a dilution process should have a negligible effect on the Neanderthal fragment lengths observed today”. I would like to see more justification of this conclusion. The effect of dilution is to decrease the frequency of Neanderthal fragments and therefore increase the frequency with which Neanderthal fragments occur in heterozygotes rather than homozygotes. This would increase the rate at which fragments were broken up by recombination and would therefore contribute to a shorter fragment length in West Eurasia. The effect may be small but the authors should address whether this effect of dilution could account for some variation in fragment lengths.

David Haig

This is a very good point. Under the SMC' model, the length of Neanderthal ancestry fragments are approximated by an Exponential distribution with parameter $rt(1-m)(1-t/4n)$ (Gravel 2012, Liang & Nielsen 2014), where m is the migration rate, r is the recombination rate and t is the admixture time. A dilution would correspond to a change of m , and even a strong dilution (e.g. from 0.03 to 0.01) would only change the fragment lengths by less than 2%. Similarly, coalescence (modelled by the $(1-t/4n)$ term) has a very minor impact on fragment lengths. Informally, this is because Neanderthal fragments are much of the time in low frequencies. We did not add any changes to the main manuscript about this because the expected effect is so small, and we would prefer not to introduce the discussion into the current narrative.

Reviewer #2 (Remarks to the Author):

By leveraging the distribution of Neanderthal ancestry fragments and patterns of mutation accumulation in Europeans and East Asians, the authors show there are significant differences in the human generation interval across non-African groups. This is an innovative idea and a very creative way to learn about historical generation time, a critical parameter in evolutionary studies. The analysis performed is compelling, but I have some concerns about the model and some results that I detail below.

Main comments

1. The key observation the authors focus on is that Europeans and East Asians differ in the % and the length of Neanderthal ancestry tracts. The author suggests the underlying model is that there was a single pulse of gene flow from Neanderthals into the ancestors of modern humans and the observed differences are due to differences in the rate of recombination clock driven by generation time. While feasible, more work needs to be done to justify this model and reject other possible reasons.

a) Single pulse of Neanderthal gene flow: While some studies support this model, there are other studies that support a model of multiple gene flow events (e.g., Vernot et al. 2016). There is also evidence for multiple pulses in ancient DNA samples such as Oase and Ust' Ishim (Moorjani et al. 2016) where the dating is more direct and based on fewer assumptions. Also given that the authors find that only half of the total Neanderthal ancestry segments are shared between East Asians and Europeans, I wonder if a model of multiple gene flow events can be rejected. Worth simulating this to test the predictions and assess whether the model of single or multiple pulses provides a better fit to the data.

We believe this is a relevant comment that is worth addressing. For that, we simulated, using msprime, whole genomes of 50 individuals for each West Eurasia (WE) and East Asia (EA) populations (similar sample sizes to the real data) in two demographic models (Figure S6):

i) One Pulse scenario in which there is a single gene flow from Neanderthals (54 kya, 2% migration rate) to the common ancestor of both populations and

ii) Two Pulses scenario, similar to the OneGeneFlow scenario with an extra Neanderthal pulse into EA dated to the last evidence of Neanderthal existence (39 kya) in order to maximize the difference in mean archaic fragment length between populations. The migration rate from Neanderthals for the second pulse is set to 0.5% ($\frac{1}{4}$ of the first Neanderthal pulse) so that the Neanderthal sequence per individual is 25% higher in EA compared to WE, as seen in the original data (Fig. 1d).

We find that the fragments of the two populations in the Two Pulses scenario are not as different between the two populations (~2.5% longer in EA) as the ones in the real data (~12% longer in EA). Furthermore, the second pulse provides EA with an excess of private sequence (58% and 49% private sequence in EA and WE respectively, Fig. S8) while in the real data we find the proportion of shared fragments to be similar between the two regions (47% and 44% private sequence in EA and WE respectively, Fig. 2a). Thus, we conclude

that an extra Neanderthal gene flow event to the EA populations does not fit with our observations.

All analyses regarding this topic are included in the Supplementary Material under section S9.

b) Differences in recombination rates across Europeans and East Asians: One key reason the authors do not consider is that the differences in recombination rate across Europeans and East Asians could underlie the differences in the Neanderthal ancestry tract lengths across these groups. There is evidence that as recombination evolves quickly due to Prdm9 and the fine scale recombination map differs between Europeans and East Asians. It would be useful if the authors can explore how changes in recombination rate would impact their results, perhaps using new population specific maps for Europeans and East Asians (e.g., Spence and Song, 2019).

We thank the reviewer for this suggestion. It is possible that if, for example, East Asians were more reliant on recombination hot spots than Western Eurasians, the same number of recombinations in East Asians would decay archaic fragments lengths at a slower rate, even though Spence and Song (2019) report a high correlation degree of recombination maps across populations.

We downloaded the recombination maps inferred by Spence and Song (2019) and computed the length of archaic fragments in recombination units (cM). We find very similar and significant differences as for physical lengths (Fig. S2). Thus, we conclude that changes in the recombination profile among populations are not the cause of our observations.

All analyses regarding this topic are included in the Supplementary Material under section S6.

c) Selection: The authors reject the possibility that selection could be driving the difference between East Asians and Europeans. However, the observation that East Asians have Neanderthal ancestry segments at higher frequency could be in part due to selection. It would be useful to perform some simulations to test models with selection— both positive and negative selection could contribute to the observed differences across populations.

While qualitatively correct, we do not believe that the quantitative effects of such selection could be appreciable. Moreover, it is unclear how one would go about specifying the selection regime on Neanderthal fragments and there would then be freedom to generate a wide range of selective models making it hard to interpret. Thus, we have refrained from following this suggestion.

On one hand, we note that prior modelling (Kelley Harris and Rasmus Nielsen 2016, Martin Petr et al 2019, Ivan Juric et al 2016) has suggested that most purifying selection on archaic fragments should happen very soon after the admixture event and thus before the separation on the Eurasian populations. On the other hand, signals of positive selection on specific fragments have been reported in the literature, such as MHC and certain skin colour genes (Fernando Racimo et al, 2015). However, the small proportion of potentially positively selected fragments, even though kept long, are unlikely to shift the mean of fragment length

distribution. If truly there were a few long fragments affecting the mean values reported due to selection, we believe that summarizing the distribution centre with the median instead should be robust to such effects. However, median patterns are very similar to what we observe with the means (Extended Fig 1a).

2. Choice of samples: The authors use SGDP dataset and compare Neanderthal ancestry patterns across West Eurasians and East Asians. However, West Eurasians themselves are a diverse group including European, Central Asian and Middle Eastern groups together – populations that likely have very different generation times and recombination rates. It would be useful to repeat the analysis with more homogenous groups. If larger sample sizes are needed then other datasets like 1000G or high coverage genomes from HGDP (Bergstrom et al. 2020) maybe useful. This will help to interpret the source of the differences more clearly.

This is a very good suggestion that will reinforce our observations and we thank the reviewer for that. We agree with the reviewer about the issues that come with grouping heterogeneous groups into superpopulations as West Eurasians. Thus, following the reviewer suggestions, we explored the mean archaic fragment length measure in more homogeneous populations offered by the HGDP dataset. In this analysis, we aimed to test how dispersed the distribution for this statistic is in homogeneous populations and attempt to replicate our observations of difference among populations from West Eurasia and East Asia. Thus, we called archaic fragments in populations of the HGDP (with representation in SGDP) which:

- i) Had the largest sample sizes for both East Asia and West Eurasia in HGDP (Palestinians = 46, Han = 33, Table S5)
- ii) A population from East Asia and another from West Eurasia that had the smallest mean length difference in our analysis of the SGDP data (Lahu and Sardinians, Fig. S4)

We find that there is a large variance in mean fragments among individuals of the more homogeneous HGDP populations (Table S5, Fig. S5). We find that the fragment length distributions of Sardinians and Palestinians are different from each other, highlighting that the West Eurasia group in SGDP is indeed heterogeneous and that the results we present thus are a weighted average of these populations. Nonetheless, we replicated the differences in fragment length between West Eurasia and East Asia regions by comparing Lahu and Sardinians. We conclude that even though our categories are made up of heterogeneous populations, the differences between regions are replicated in the independent HGDP data.

All analyses regarding this topic are included in the Supplementary Material under section S7.

3. Inconsistency in signals across populations

The authors show that decrease in Neanderthal ancestry tract length is correlated to increase in derived alleles in Europeans. This makes sense. However, South Asians have similar % of derived allele as West Eurasians but fragment lengths are longer (Table S3 and S6). This is contradictory to their model predictions. Could there any technical issue impacting this result?

It is correct that the number of derived alleles in South Asians does not follow the qualitative pattern that we predict. We see a couple of potential reasons for that. First, South Asians have ancestries from both West Eurasia and East Asia to varying extent. Therefore, Denisovan ancestry might be a reason why the fragments tend to be longer in that population. Second, the accumulation of derived allele measure is underpowered to establish with confidence if West Eurasians have more or less derived alleles than South Asians with the sample sizes in hand.

4. Figure 3 - For each mutation type, the authors show slopes for two models - correlation in derived allele accumulation and Neanderthal ancestry fragment length and parental age effects. They suggest there is a strong correspondence in the slopes across the two models. However, this analysis is very qualitative and in particular, these slopes look fairly different for many mutation types. It would be useful if the authors can provide quantitative estimates of the similarities and differences for each mutation type.

In order to get a quantitative estimate, we have correlated the slopes for each dataset (Fig. S11) and obtained a very good correlation between estimates (Linear model : deCODE slope = $-1.196e-05 + 1.058(\text{SGDP slope})$. P value = $1.768e-3$. Adjusted $R^2 = 0.7414$). This is now included in Supplementary Materials section S12 and alluded to in the main text.

5. TCC->TCT

It is interesting that the authors find that differences in generation time contribute to the increase in TCC->TCT mutations in Europeans. Could the authors provide some quantitative estimates of what proportion of this difference is explained by generation time vs. other factors?

We already in the discussion report the estimate that 55% of the variance of the TCC>TTC mutation among individuals can be explained by differences in generation time, here approximated by the mean archaic fragment length. We believe this is a great proportion of the variation since mutation spectra can be influenced by other factors - such as GC biased gene conversion - determining the abundance of each mutation type. Furthermore, as seen in the trio data, the correlation between parental age at conception and TCC>TTC mutation proportion is noisy. We have now stated this more clearly in the discussion.

6. Direct estimate of generation time

The authors provide indirect estimates of generation time differences between Europeans and East Asians. Could they relate the N fragment length distributions and mutation patterns to provide a direct estimate of generation time?

This would be a great thing to do but, at present, we think it would be associated with too much uncertainty. For fragment lengths distribution, it will depend on the exact time of Neanderthal admixture, and whether it was instant over time or a prolonged meeting. Furthermore, we would have to model the false positive and false negative fragments discovered by the model used in the study. For mutational patterns, we are comparing derived allele accumulation with de novo mutations, but the relationship between these are not completely understood yet. For example, mutations between strong and weak base pairs are affected by biased gene conversion, thus, given enough evolutionary time, this force will affect the relative proportions of mutations increasing the frequency of C and G. With more

trios sequenced in the future, including sequencing of non-European trios, together with archaic fragment distributions from many thousand individuals, it should be possible to directly model this and we would be interested in doing so.

7. Sex-specific changes in generation time

The authors use the enrichment of C>G mutations to compare the maternal generation time across populations. This is an interesting idea. However, the analysis seems very tentative as the ratio seems fairly dispersed within and across populations. Moreover, X/A is impacted by the sex-specific generation time though the Neanderthal ancestry tract lengths on the autosomes depend on the sex-averaged generation time. Further, some of the populations like Americas that show an increased ratio are admixed (with Native American, European or African ancestry) and so it is not clear how to interpret these results -- which ancestry group has a long maternal generation time. A more clear quantitative model is needed here to justify the conclusions.

We agree with the reviewer that these results are very tentative and not very conclusive with the amount of data at hand. We have changed the main text such that it is more clear that we are not drawing any firm conclusions.

Minor comments:

1. Specificity and sensitivity of Neanderthal ancestry detection HMM. The authors state that this method has lower sensitivity to detect shorter segments. Could this impact the differences across populations? Would be useful if authors can provide information about the specificity and sensitivity of the method as a function of ancestry tract lengths.

We have previously considered this question in the Skov et al. 2018 (SI figure 2). In the present analyses, smaller sensitivity for smaller fragments is conservative against the differences we report, since we are more likely to miss short fragments in the populations where the fragments are already short.

2. On page 2 of results, authors claim that after excluding Denisova fragments the total sequence covered by archaic fragments is almost identical. Does this not suggest that the 7% difference seen earlier is driven by Denisovans in large part?

Yes, this is what we suggest and it is consistent with the previous reports of the amount of Denisovan contributions to East Asians (around 0.2%). We now make this more clear in the main text.

3. The authors use parental age effects estimated in Europeans and compare them to N fragments across populations. It would be useful if the authors can mention this point in the discussion or results, as these mutation parameters and parental age effects may themselves differ across human populations.

This is true and we have followed the suggestion by clearly stating it in the manuscript.

4. Difference in generation time.

The difference in generation time of 5 years between Europeans and East Asians seems tenuous and depends on the timing of the OOA migration. This has been estimated to be

between 50,000-100,000 years and is closer to 100,000 based on the pedigree mutation rate as used for other analysis. See Schiffels & Durbin 2014 or Spiedel et al. 2020.

The reviewer is right on the fact that our estimate on the generation time difference between the two populations is highly subjected to the Out of Africa migration (split between Africans and non-Africans) - here assumed to be 60,000 years ago - and the split between West Eurasians and East Asians - here assumed to be 40,000 years ago - in our oversimplified model. As discussed in the papers cited, the Out of Africa event does not seem to be a discrete event, but rather a prolonged demographic episode.

As we state in the main text, we are counting derived alleles that were accumulated after the Out-of-Africa, but consider that the difference in this quantity between genomes from West Eurasia and East Asia could only happen while their ancestral populations were apart. Therefore, we need to assume dates. The ratio between the two rates assumed ($60,000/40,000 = 1.5$), gives us the correction that we apply to the derived allele accumulation rate. Thus, if we push further back the Out-of-Africa event, we are going to increase the correction factor, resolving to a greater difference in generation intervals between the two. That is why we believe that the dates assumed lead to more conservative estimates, i.e. smaller differences in inferred derived allele accumulation and thus in estimated generation intervals. All these calculations are explained in SI 11.

Reviewer #3 (Remarks to the Author):

This article aims at inferring ancient socio-biological patterns (age of mothers and fathers) using genetic data of 2 two types : Neandertal inherited fragment length (that provide insights on the recombination rate) and number of derived alleles (that depends on the number of de novo mutations).

The article is well-written and provides convincing arguments. It also enlightens the process of admixture between Sapiens and Neandertal. However, I have two major concerns.

1) The authors use the mean archaic fragment length of 3 ancient genomes to infer the differences in the number of generations since their split between Asian and European (l. 133-l. 142). However, these genomes are either very ancient (Ust-Ishim, 45 ky cal BP) or quite recent (Stuttgart 7 ky cal BP and Loschbourg 8ky cal BP), thus providing a poor estimation of the decrease of the fragment length through time. I suggest that the authors add a few genomes of west- Eurasian ancestry to their data set, such as Kostenki14 (coverage: 16.1 X, age : 37 ky cal BP) ; Kotias (coverage : 12.2 X, age : 9.5 ky cal BP) or Sunghir (coverage : 10.75 X, age : 34 ky cal BP). It would also be interesting to compare these ancient genomes to Anzick-1, from the Clovis Culture in North America (coverage: 14.4 X, age : 12.5 ky cal BP), or to Yana1 and Kolyma 1 from paleolithic Siberian population (coverage respectively : 25 X and 14.3X, age respectively : 9.7 ky cal BP and 31.6 ky cal BP). The authors should also take into account their lack of precision in the estimation of the “missing recombinations” in Asian and moderate their conclusions (l. 138 – l.142)

We thank the reviewer for this suggestion. We have followed that by adding Kolyma, Yana1 and Sunghir3 to the analysis as well as performed a reanalysis of the previous genomes in order to have similar SNP calling. We also analysed the Anzick genome data but concluded that the transition/transversion ratio indicates an excess of SNPs due to ancient DNA damage. The results are now presented in Figure 2b and in the Supplementary Information S3 and S5. As expected these ancient genomes add to the picture a gradual decay in archaic fragment lengths over time and support our suggestion that East Asians have had fewer generations of such decay than the West Eurasian population. The main text and Supplement sections have been changed to include these new results.

2) On the l. 200-l. 201, the authors suggest that archaic fragment length and derived alleles accumulation are not related. On the contrary, mutation rate depends also on the GC-biased conversion during recombination. This phenomenon increases the fixation of C and G alleles compared to T and A. GC-biased gene conversion is linked in human to the activity of PRDM9 protein. Different alleles of PRDM9 exist in the human species and some affect recombination rate (for instance between European and African : <https://advances.sciencemag.org/content/5/10/eaaw9206>). I wonder if this phenomenon could affect the authors results on recombination and mutation rate.

This is an interesting thought. First of all, we would like to point out that we consider the derived allele accumulation analysis to be independent from the Neandertal fragment length analysis in the sense that we are assessing different parts of the genome: regions

with vs without Neanderthal fragments. Thus, Neanderthal introgression will not influence the counts of alleles accumulated or the spectrum.

We agree with the reviewer that recombination has a mutagenic effect and also modifies the mutation spectrum, for example, via GC-biased gene conversion. However, recombination represents a small fraction of the global mutation rate per generation, likely less than 5% (Halldorsson et al 2019). Furthermore, in the analyses of derived allele accumulation, we explore Neanderthal-free regions of the genome, which will have smaller recombination rates (Skov et al 2020, Schumer et al 2018), diminishing the effects pointed out by the reviewer. Finally, if the recombination event in an archaic fragment did cause a mutation, this mutation would likely be counted as part of the filtered archaic fragment since it would be a variant not seen in the unadmixed outgroup.

In conclusion, we trust that the inferences from archaic fragments and derived allele accumulation in non-archaic regions are very close to be independent measures.

I have a few other observations

3) The authors observe that the Skov et al. method has difficulties to identify very short fragments (l. 100). What is the threshold? could that change the mean/median length of the fragments for both populations (for instance with a higher number of short fragments in Asian genomes)?

We have previously considered this question in the Skov et al. 2018 (SI figure 2) where we show that our HMM performs poorly at capturing fragments less than 40 Kb.

In the present analyses, smaller sensitivity for smaller fragments is conservative against the differences we report, since we are more likely to miss short fragments in the populations where the fragments are already short.

4) The authors suggest that only one admixture event occurs in the European/East Asian ancestors (l. 111-112). However, the homogeneity of Neanderthal alleles in European and East Asian genomes do not rule out several admixture events with only one homogeneous Neanderthal population. Indeed, it seems that Neanderthal groups were highly homogeneous on a genetic point of view. If ancestors of East Asian populations have experienced more admixture events/more recent admixture event with this Neanderthal population, it could explain the higher percentage of Neanderthal alleles and the longer Neanderthal fragments in East Asian genomes.

We want to clarify that when we compare Neanderthal sequence in multiple populations we refer to the fragment overlapping rather than differences among fragments in nucleotide identity. Our main assumption has been that a more recent Neanderthal gene flow private to East Asians would provide Neanderthal fragments in new genomic positions and thus a larger proportion of private genomic positions of archaic fragments in East Asians, a pattern not seen in the data.

We now explicitly demonstrate this verbal argument through simulations (Supplementary Information S9). We find that our observations are not compatible with a second Neanderthal admixture to East Asians since: 1) this event does not create big differences in the mean

archaic fragment length and 2) increases the proportion of private archaic fragments in East Asians compared to West Eurasians.

5) In line with others, the authors consider that Neandertal ancestry may have been diluted by gene flow from Basal Eurasian with little/no archaic alleles. In the Skov et al. 2018 paper, it is suggested that “We note that other types of population structure, for example involving continual gene flow, could also create signals under our model.” Could this gene flow impair identification of archaic fragments?

The method classifies the genome into windows of archaic and non-archaic origin. It is able to do that because polymorphisms shared with an unadmixed outgroup more closely related to the ingroup than the archaics - in this case, African populations - are removed from the genomes of the samples in which we want to detect admixture - the ingroup, in this case non-African populations from SGDP -. By doing that, fragments that come from the archaic population will be highly SNP dense compared to fragments from a human origin, from which lots of polymorphisms had been removed.

Then, if there was gene flow from a Basal Eurasian population to either of the populations assessed in the study, the discovery rates of the model should not be affected. This is because Basal Eurasians are thought to be a sister group of the common ancestors of Eurasians compared to Africans; thus, Basal Eurasians would be considered as ingroup and Africans would still serve as outgroup. Therefore, even though West Eurasians received gene flow from Basal Eurasians, and even if this was extended through time, filtering out African alleles from West Eurasians would have the same effect as to East Asians.

6) In the Fig.4, it seems that East Asian populations have a higher paternal age at conception than other Asian populations (Fig.4.a) AND a higher maternal age than Central Asian/Siberian population. But this difference does not reflect on the mean Neandertal fragment length. Or, to rephrase, Central Asian/Siberian populations do not appear significantly different from West Eurasian populations for paternal and maternal age but have significant Neandertal fragment length. How do you explain this?

The X/A ratio gives us information about the relative difference between the paternal and maternal age at conception in a population. Thus, although two populations might have had different mean parental age - measured with the mean archaic fragment length -, the ratio between their parental ages might be similar (for example, if Central Asia and Siberians had mothers and fathers both having descendants at 29 years and South Asians at 25 years). It is true that for the same mean parental age, greater values of X-to-A ratio can be interpreted as older fathers and/or younger mothers (for example, the case for East Asians and Central Asians and Siberians). Compared to this statistic, The C>G abundance, instead, is directly proportional to the maternal age and does not depend on both parental ages.

Related to the reviewer's concerns, we first want to point out that we haven't modeled directly the quantitative relationship between the increase or decrease of a parental age with the variation of the signals analyzed. For example, there is evidence that the C>G enrichment in older mothers might not be linear, but instead quadratic (Gao Z et al, 2019). Furthermore, how the same mean paternal and maternal ages perturbation affects each statistic at the same time is also not understood yet. Thus, it is not possible to give an

accurate prediction of the and compare them among populations. Furthermore, we believe that we would also need more data in order to get better estimates for such noisy statistics and formulate stronger conclusions. Also, more data will help us to study more homogeneous populations since here we pull populations with different demographic histories into broad categories.

Nonetheless, we think there is potential for these signals to be investigated further in future studies to disentangle all these questions. That is why we give interpretation in broad-strokes of the results obtained.

7) Last remark: the authors write that this difference in the generation time may be explain by agriculture that decrease generation time (in Western Eurasian). But this difference is already seen in Stuttgart and Loschbour (just before/after Neolithic) ; and the onset of agriculture is more or less the same between East Asian and Western Eurasia. Thus, I do not think this can be the cause of the difference.

We do not intend to imply that farming is the sole explanation for the differences observed and also caution about this in the discussion. The text has been changed to make this more clear.

8) Overall, the text is carefully written and pleasant to read. Figures are very clear, useful and beautifully done. My main point on the wording is that the aim of the article does not appear clearly at the beginning: both the introduction and the abstract are focused on the admixture between Neandertal and Sapiens, which is in fact only a tool for studying a Sapiens-related phenomenon.

Thank you for the nice comments. We now introduce the idea that archaic fragment length might teach us about human demographic processes earlier in both the abstract and in the introduction.

9) In the bibliography, there is a few typos for the following articles: (2) Bergström et al. ; (9) Skov et al. ; (16) Fu et al. ; (27) Fenner et al.
Also a typo l. 150 “non Afrcan populations”

Thank you for pointing out these typos. We have incorporated the changes accordingly.

To conclude, I consider this paper as very interesting and worth of Nature Communication, since the comments and concerns I expose are correctly addressed.

REVIEWERS' COMMENTS

Reviewer #1 (Remarks to the Author):

My one substantive point in the first round was that dilution in West Asia would decrease the frequency of Neanderthal fragments. Therefore increasing the frequency of heterozygotes and the frequency with which fragments would be subject to detectable recombination. The authors kindly consider this "a very good point" but choose not to address it because the effect would be small. I agree. If the frequency of Neanderthal segments then the frequency of homozygotes will be very small and the effect negligible (under the assumptions of the fitted model. The model, I presume, assumes a panmictic population. If one modeled a metapopulation with allele frequency variation among demes, the frequency of homozygotes would increase but probably not enough to make a significant difference.

All models make simplifying assumptions. I consider the hypothesis that the different fragment lengths in East and West Eurasia is due to a difference in generation length to be a hypothesis worth publishing.

Reviewer #2 (Remarks to the Author):

The authors have addressed most of the technical concerns raised. The manuscript is much improved and the interpretation is clearer. I have some minor comments about adding some qualifications to improve the interpretation further and typos that need to be fixed.

1. Abstract

"Altogether, our results suggest consistent differences in the generation interval across Eurasia, by up 10-20%, over the past 40,000 years."

Change to:

"Altogether, our results suggest consistent differences in the generation interval across Eurasia, by up 10-20%, since the split of East Asians and West Eurasians over the past 40,000 years."

2. Results

a. Very similar and significant differences are also found in the independent Human Genome Diversity Project (HGDP) data² when comparing more homogeneous populations from each region (Sardinians and Lahu, S7, Data3_HGDParchaicfragments.txt).

Please add “though there is large variation in patterns within continental groups observed”. In fact for some populations, the patterns in East Asians and West Eurasians are similar in S7.

b. Comparison of recombination rates across groups

At fine-scales, population specific maps have high error rates as quantified in previous studies (e.g., Sankararaman et al. 2008, Lipson et al. 2016 and Moorjani et al. 2016) and seen in the supplementary analysis (e.g. longer map length in East Asians). These errors are not uniform and may differ by population and time. Worth adding –

“Assuming the similar accuracy and resolution in both genetic maps, we find that the differences between Western Eurasia and East Asia are quantitatively very similar to those for the fragments measured in base pairs (Fig. S2, S6).”

c. Correlation in slopes for

$\text{lm}(\text{mutation_fraction} \sim \text{mean_fragment_length})$

$\text{lm}(\text{mutation_fraction} \sim \text{mean_parental_age}, \text{weights} = \text{n_probands})$

It would be useful to include more details for this analysis since its not clear what was actually done. Are you plotting the regression coefficients for the dependent variables in this case against each other?

d. “However, we acknowledge that the amount of data is still limited for this test and the conclusion is thus preliminary.”

In addition to data, the models used here do account for many factors like sex bias gene flow, variance in reproductive success etc. and so perhaps useful to say

“However, we acknowledge that the amount of data is still limited and other factors like sex bias gene flow or variance in reproductive success may also contribute and the conclusion is thus preliminary.”

e. Are the effect sizes of differences in generation times of males and females consistent with the expectation from autosomes? Would be useful to comment.

f. Typos - Please check supplement for typos. e.g., "values" is misspelled in a number of tables, population names are also misspelled in a couple of places, etc.

Reviewer #3 (Remarks to the Author):

The authors answered with precisions to all the questions I have asked in the first round of reviews. They performed supplementary analyses that are convincing and amended the manuscript as necessary.

As a consequence, I consider that the manuscript can be published as is, only for a small modification :

In the Discussion (p9 ; l 284-287), the authors suggest that agriculture could have had an effect on generation time. But,

- 1) Western Europe and Eastern Asia turned to agricultural way of life approximately at the same time
- 2) the differences in the generation time is seen as early as Stuttgart genome, whose switch to farming and breeding date back only a few generations ago.

So, I do not think that such an hypothesis is sustained by the data. I would remove this sentence.

This considered, this is a brilliant article, and look forward seeing it published.

REVIEWERS' COMMENTS

Reviewer #1 (Remarks to the Author):

My one substantive point in the first round was that dilution in West Asia would decrease the frequency of Neanderthal fragments. Therefore increasing the frequency of heterozygotes and the frequency with which fragments would be subject to detectable recombination. The authors kindly consider this "a very good point" but choose not to address it because the effect would be small. I agree. If the frequency of Neanderthal segments then the frequency of homozygotes will be very small and the effect negligible (under the assumptions of the fitted model. The model, I presume, assumes a panmictic population. If one modeled a metapopulation with allele frequency variation among demes, the frequency of homozygotes would increase but probably not enough to make a significant difference.

All models make simplifying assumptions. I consider the hypothesis that the different fragment lengths in East and West Eurasia is due to a difference in generation length to be a hypothesis worth publishing.

We agree with the reviewer's observations. Thank you for such comments.

Reviewer #2 (Remarks to the Author):

The authors have addressed most of the technical concerns raised. The manuscript is much improved and the interpretation is clearer. I have some minor comments about adding some qualifications to improve the interpretation further and typos that need to be fixed.

1. Abstract

"Altogether, our results suggest consistent differences in the generation interval across Eurasia, by up 10-20%, over the past 40,000 years."

Change to:

"Altogether, our results suggest consistent differences in the generation interval across Eurasia, by up 10-20%, since the split of East Asians and West Eurasians over the past 40,000 years."

This is a good suggestion since it might not be obvious for the reader that 40,000 years correspond to the split time between Eurasian populations. The limiting length of the abstract (200 words) forces us to cut down the length of such a section, and thus we do not include the suggested sentence in the abstract. Nonetheless, we included a much-clarifying sentence in the introduction.

2. Results

a. Very similar and significant differences are also found in the independent Human Genome Diversity Project (HGDP) data2 when comparing more homogeneous populations from each region (Sardinians and Lahu, S7, Data3_HGDParchaicfragments.txt).

Please add “though there is large variation in patterns within continental groups observed”. In fact for some populations, the patterns in East Asians and West Eurasians are similar in S7.

That is right. Unexpectedly, we find that the mean archaic fragment length presents an extensive variation in each population for the HGDP dataset. It might be worth trying other summary statistics in order to describe the archaic fragment length distributions in future iterations of the project. We now include a sentence in the results section that mentioning this fact.

b. Comparison of recombination rates across groups

At fine-scales, population specific maps have high error rates as quantified in previous studies (e.g., Sankararaman et al. 2008, Lipson et al. 2016 and Moorjani et al. 2016) and seen in the supplementary analysis (e.g. longer map length in East Asians). These errors are not uniform and may differ by population and time. Worth adding –

“Assuming the similar accuracy and resolution in both genetic maps, we find that the differences between Western Eurasia and East Asia are quantitatively very similar to those for the fragments measured in base pairs (Fig. S2, S6).”

We believe that adding the “Assuming the similar accuracy and resolution in both genetic maps” disclaimer can be either interpreted as lack of confidence on the genetic maps used or the analysis performed. Any of the two would invalidate the results obtained. Furthermore, in the supplementary information we caution inconsistencies that we find such as the difference in map length. Therefore, we choose to not add the suggested sentence and let the reader judge the validity of our conclusions.

c. Correlation in slopes for

$\text{lm}(\text{mutation_fraction} \sim \text{mean_fragment_length})$

$\text{lm}(\text{mutation_fraction} \sim \text{mean_parental_age}, \text{weights} = \text{n_proband})$

It would be useful to include more details for this analysis since its not clear what was actually done. Are you plotting the regression coefficients for the dependent variables in this case against each other?

We now more explicitly explained the analysis in the main text.

d. “However, we acknowledge that the amount of data is still limited for this test and the conclusion is thus preliminary.”

In addition to data, the models used here do account for many factors like sex bias gene flow, variance in reproductive success etc. and so perhaps useful to say

“However, we acknowledge that the amount of data is still limited and other factors like sex bias gene flow or variance in reproductive success may also contribute and the conclusion is thus preliminary.”

We certainly agree with this comment. This is why we note this in lines between 231 and 234:

“Thus, an increase of the relative male-to-female generation interval is expected to increase the X chromosome to autosomes (X-to-A) mutation accumulation ratio³³, although other factors such as reproductive variance and changes in population size can also influence the ratio.”

We think it is not necessary to repeat such information in the discussion.

e. Are the effect sizes of differences in generation times of males and females consistent with the expectation from autosomes? Would be useful to comment.

We agree with this. In the main text we attempt to relate the results obtained from the sex-specific mutation patterns with what is observed in the general pattern. For example, between lines 252 - 258:

“Notably, America has a higher ratio than Central Asia and Siberians for similar Neanderthal fragment lengths, suggesting a relatively larger impact of old mothers to the overall mean generation time throughout their history. This is in line with the X chromosome analysis in that longer generation times in America were more driven by older mothers as compared to older fathers in East Asia with an intermediate increase of both parental ages in Central Asia and Siberia.”

f. Typos - Please check supplement for typos. e.g., "values" is misspelled in a number of tables, population names are also misspelled in a couple of places, etc.

Thank you for pointing out these orthographical mistakes. We have now corrected those mistakes.

Reviewer #3 (Remarks to the Author):

The authors answered with precisions to all the questions I have asked in the first round of reviews. They performed supplementary analyses that are convincing and amended the manuscript as necessary.

As a consequence, I consider that the manuscript can be published as is, only for a small modification :

In the Discussion (p9 ; l 284-287), the authors suggest that agriculture could have had an effect on generation time. But,

1) Western Europe and Eastern Asia turned to agricultural way of life approximately at the same time

2) the differences in the generation time is seen as early as Stuttgart genome, whose switch to farming and breeding date back only a few generations ago.

So, I do not think that such an hypothesis is sustained by the data. I would remove this sentence.

This considered, this is a brilliant article, and look forward seeing it published.

The purpose of the original sentence was to open a debate about which factors could have played a role in changing the generation time in modern humans, directly suggesting specific events. In accordance to the reviewer comment, we have now tunned the sentence down and suggesting instead more general causes. We appreciate the reviewer comments and kind words.